# Validity and Reliability of International Physical Activity Questionnaires for Adults across EU Countries: Systematic Review and Meta Analysis

**DOI:** 10.3390/ijerph17197161

**Published:** 2020-09-30

**Authors:** Vedrana Sember, Kaja Meh, Maroje Sorić, Gregor Starc, Paulo Rocha, Gregor Jurak

**Affiliations:** 1Faculty of Sports, University of Ljubljana, 1000 Ljubljana, Slovenia; kaja.meh@fsp.uni-lj.si (K.M.); maroje.soric@kif.unizg.hr (M.S.); gregor.starc@fsp.uni-lj.si (G.S.); gregor.jurak@fsp.uni-lj.si (G.J.); 2Faculty of Kinesiology, University of Zagreb, 10110 Zagreb, Croatia; 3Portuguese Institute of Sport and Youth, 1250-190 Lisbon, Portugal; paulo.rocha@ipdj.pt

**Keywords:** measurement characteristics, policy, European Union, measurement properties, language version, IPAQ, GPAQ, EHIS-PAQ

## Abstract

This review and meta-analysis (PROSPERO registration number: CRD42020138845) critically evaluates test-retest reliability, concurrent validity and criterion validity of different physical activity (PA) levels of three most commonly used international PA questionnaires (PAQs) in official language versions of European Union (EU): International Physical Activity Questionnaire (IPAQ-SF), Global Physical Activity Questionnaire (GPAQ), and European Health Interview Survey-Physical Activity Questionnaire (EHIS-PAQ). In total, 1749 abstracts were screened, 287 full-text articles were identified as relevant to the study objectives, and 20 studies were included. The studies’ results and quality were evaluated using the Quality Assessment of Physical Activity Questionnaires checklist. Results indicate that only ten EU countries validated official language versions of selected PAQs. A meta-analysis revealed that assessment of moderate-to-vigorous PA (MVPA) is the most relevant PA level outcome, since no publication bias in any of measurement properties was detected while test-retest reliability was moderately high (r_w_ = 0.74), moderate for the criterion (r_w_ = 0.41) and moderately-high for concurrent validity (r_w_ = 0.72). Reporting of methods and results of the studies was poor, with an overall moderate risk of bias with a total score of 0.43. In conclusion, where only self-reporting of PA is feasible, assessment of MVPA with selected PAQs in EU adult populations is recommended.

## 1. Introduction

Increasing the level of physical activity (PA) has become one of the priorities of public health policies in most developed countries in the world [1]. Over the last thirty years, we have witnessed an accelerated increase in the quantity of interventions to increase PA worldwide, although with limited effects [2,3,4,5]. Creating optimal policies and planning effective interventions aimed at increasing PA is not possible without reliable data on the prevalence of physical inactivity [1]. Hence, numerous global authorities have called for concerted efforts in PA surveillance [6,7,8]. Conversely, how to execute PA monitoring is not entirely clear. Although methods for the assessment of PA are numerous, given the complex nature of PA, none of the currently available methods can assess all PA dimensions (duration, frequency, intensity and type of PA).

Based on the literature review, we can classify scientific methods for determining PA as direct observations or objectively assessed PA and indirect or subjectively assessed PA [9,10,11]. Large PA surveillance systems have, until recently, relied solely on PA questionnaires (PAQs) as one of the subjective assessments of PA [12]. Questionnaires are easy to apply in large groups of individuals and are therefore the basic method of assessing PA in large epidemiological studies. However, this method is subject to recall bias, which typically leads to overestimation of PA [13]. Therefore, some of the large PA surveillance systems have recently begun to rely on objective assessments by accelerometers to monitor activity levels [14]. Although the validity of accelerometers has been tested in numerous settings [15,16,17] and despite the fact that accelerometers have proved to be more reliable and valid than PA questionnaires [18,19,20], several shortcomings have to be noted, such as the underestimation of energy expenditure during uphill walking, cycling, load carrying, etc. [21] Additionally, other important issues for large surveillance systems might be costs [22], demanding data reduction procedures and obtrusiveness of devices, which reduces compliance and increases non-wear time [23], specialized training required for assessors and the need for the physical proximity of participants. On the other hand, the advancement in technology has led to the development of commercial activity monitors for personal use. Recent evidence on accuracy of these devices indicates that this technology could be a very useful tool for surveillance systems [15,24,25,26,27,28,29,30]; however, at the moment, PAQs still prevail [12,31,32].

In designing a monitoring system for PA, a harmonized approach using a single, international instrument is preferred to enable cross-country comparisons. However, because PA is a behavior, the cultural environment should be taken into account when the same PA questionnaire is used in different countries [1,33]. Namely, most PAQs rely on a person remembering activities they participated in, or self-estimates of the intensity of the recalled PA [34]. Therefore, the cultural context and country-specific types of PA are very important for the interpretation of questions, and consequently for the content validity of a PAQ [33,35]. 

Within the project EUPASMOS, which aims to establish PA, sedentary behavior patterns and sport participation monitoring framework in the European Union (EU) member states, we searched for studies performed in the EU, and described measurement characteristics of nationally adapted versions of the three most commonly used international PAQs intended for trans-national surveillance and aimed at generating comparable estimates across countries: (i) International Physical Activity Questionnaire-Short form (IPAQ-SF), which was the first instrument developed for PA surveillance activities, implemented in several large surveillance programs both globally and in Europe [36], and is the most frequently used and validated PAQ [37,38]; moreover, items from this PAQ are included in Eurobarometer, which is one of the tools used for decision-making in the EU [39] and is also the most commonly used PAQ in European national surveillance systems [40]; the (ii) Global Physical Activity Questionnaire (GPAQ) was designed by the World Health Organization (WHO) as a part of the STEPwise approach to chronic disease risk factor surveillance and was implemented in more than 120 countries globally [35,41], and is the most widely used PAQ also internationally [40]; and (iii) European Health Interview Survey-Physical Activity Questionnaire (EHIS-PAQ), created under the auspicies of Eurostat [40], and is used in the only currently available EU-wide surveillance system of all member states, and includes PA [33,42].

Selected PAQs have some common features, but many specifics. IPAQ is an instrument that was developed to establish a standardized and culturally adaptable measurement tool for measuring PA in different cultural areas of the world [33]. The short form of IPAQ (IPAQ-SF) comprises nine items [35]. IPAQ-SF is an open-ended questionnaire, last 7-days recall, available in English and many other languages, covering four domains of PA (leisure time PA, domestic activities, work-related PA, transport-related PA) in each of four types of PA (sitting, walking, moderate-intensity activities and vigorous-intensity activities) [43]. The outcome of the IPAQ-SF is MET min/week and PA category score. Although the original version of the IPAQ (IPAQ-L) is slightly more reliable, it has proven to be too long and less comprehensible compared to IPAQ-SF [44], making the latter more user-friendly. GPAQ uses a typical week recall and is somewhat longer than the IPAQ-SF. It has 16 questions and covers three domains of PA (work, transport and leisure) and sedentary behavior [45]. GPAQ can differentiate between two intensities of PA (vigorous and moderate) [35]. Both GPAQ and IPAQ were designed to compare PA levels in different cultural settings around the world. On the other hand, EHIS-PAQ is an EU-specific questionnaire within the European Health Interview Survey. EHIS-PAQ is a domain-specific questionnaire with last 30-days recall, which includes 8 questions, covering three domains of PA (work-related, transport-related and leisure time), and distinguishes between aerobic and muscle-strengthening PA [46]. Although some reviews and meta-analysis of measurement properties of PAQs have already been published [38,47,48,49], there is still lack of knowledge addressing this issue on the European population is very multi-national, multi-cultural and multi-lingual. 

Therefore, the purpose of this systematic review and meta-analysis is to critically appraise, compare and summarize the measurement properties (reliability, criterion validity, construct validity) of PAQs most commonly used in trans-national surveillance systems for adults in EU-official language versions, taking the methodological quality of these studies, as well as the quality of the evidence, into account.

## 2. Materials and Methods 

The meta-analysis was performed and reported in accordance with the Preferred Reporting Items for Systematic Reviews and Meta-Analysis (PRISMA) guidelines [50,51]. The present work was registered at the International Prospective Register for Systematic Reviews, identification code CRD42020138845.

### 2.1. Search Strategy 

An identical search strategy was employed in PubMed, SportDiscus, Scopus, Dart and ResearhGate databases, looking for studies describing measurement properties of three international PAQs from April to May 2018. The search was later updated to include articles published between May 2018 and May 2020. We used the following search string “name of the questionnaire e.g., IPAQ AND (valid * OR reliab * OR repeat * OR reproducib * OR assess * OR measure *)”. Additional studies were identified by manually searching the reference lists of the full papers identified during the search. Grey literature was additionally reviewed through ResearchGate, Google Scholar and Mendeley, using only keyword “name of the questionnaire e.g., IPAQ AND valid *” and through personal communication of members of the research team with other scientists. Additional literature that corresponds to the eligibility criteria of the present review was also obtained through an online questionnaire posted on the platform 1KA (University of Ljubljana, Faculty of Social Sciences) with the help of the World Health Organization within EUPASMOS project activities. National health-enhancing physical activity (HEPA) focal points were asked to report on any national research, reports and doctoral theses, published in their national languages that examined the measurement properties of any of the three PAQs included in this study. All articles generated from the initial search were stored on Mendeley reference management software and researcher network (Elsevier, Amsterdam, The Netherlands) which was used to remove duplicate references.

### 2.2. Eligibility Criteria 

Studies included in the present review had to be peer-reviewed, include healthy adults (18 years old or older), carried out in one of the EU countries (28 countries included—United Kingdom was still part of the EU and was, therefore, included in this review) and published in one of the EU’s 24 official languages. For the purposes of the present review only those studies which examined one or more of the most commonly used standardized PAQs in the EU [35,36,37,38,39,40,41], were included: IPAQ-SF, which was the first developed PA surveillance instrument [36] and the most frequently used PAQ in EU [37,38]; GPAQ, which is with 120 countries is the most used PAQ in the world [35,40,41]; and EHIS-PAQ, which is the only available EU surveillance system used by all EU member countries [33,42]. Studies needed to report the following characteristics: (i) PAQ translation protocol, (ii) mode of administration (interview, self-administered) and (iii) reliability or (iv) concurrent validity or (v) criterion validity of included PAQ. Studies performed in special populations (e.g., participants with specific medical conditions) were excluded.

The time interval between the test and retest must have been described and short enough that the subject’s PA could not have changed, but long enough to prevent recall [37]. For PA assessment during the current or previous week, a recall period of 1 day to 3 months was considered appropriate [37].

### 2.3. Quality and Risk of Bias Assessment

The assessment of the risk of bias of included studies was conducted using the criteria, previously used by Sneck [52] and Sember [53], which includes the criterion of power calculations. Each study received “0” (does not meet the criterion) or “1” (meets the criterion) for each criterion based on an analysis of the reporting in the original article. Methodological quality was assessed following the QAPAQ checklist [54], which was developed specifically for qualitative assessment of PA questionnaires. Risk of bias assessment and methodological quality was performed by two independent reviewers (Vedrana Sember and Kaja Meh)

### 2.4. Data Extraction and Statistical Analysis 

Abstract and full-text article screening, data extraction and quality assessment were performed by two independent reviewers (Vedrana Sember and Kaja Meh) who also checked all databases and identified potential studies through the search process to identify potentially relevant articles. In case of uncertainty, a third and fourth reviewer (Gregor Jurak and Gregor Starc) screened the article. Summary tables of entered data were checked with the trial protocol and latest trial report or publication. Any discrepancies or unusual patterns were checked with the study principal investigator. A Hunter-Schmidt estimate was used for reducing the amount of bias and Fisher’s z transformation was applied to samples’ correlations to display publication bias [50,51]. We also assessed publication bias with Egger’s bias test [55] for all PA constructs, separately for reliability, concurrent and criterion validity.

For further analysis, correlation (r_w_) coefficients were determined by the Hunter-Schmidt approach [55,56], which was multiplied by the sample size of each study (r_w_ × N). The generalizability of r_p_ was corrected using an artefact correction and variance sample. For weighted means (r_w_), 95% credibility interval: CI_w_ = r_w_ + 1.96√V_p_ and I^2^ and Q statistics to measure heterogeneity of ES were calculated. Statistical analysis is explained in more detail elsewhere [53]. A forest plot was generated with online software “DistillerSR Forest Plot Generator” from Evidence Partners.

### 2.5. Data Synthesis 

Results of 20 studies were synthesized into four categories: (1) General characteristics of selected studies of PAQs across the EU; (2) reliability of PAQs in selected studies across the EU; and (3) concurrent validity of PAQs in selected studies across the EU: Criterion validity of PAQs in selected studies across EU. The systematic review synthesized 20 studies and the meta-analysis synthesized only 17 studies, since it was performed only for moderate (MPA), moderate-to-vigorous (MVPA), vigorous (VPA) and total PA (tPA), and 3 studies failed to report these metrics.

### 2.6. Grading the Level of Evidence

Reliability levels of evidence were formulated following van Poppel and colleagues (2010) levels of evidence: (1) adequate time between test and retest and use of interclass correlation (ICC), Kappa or Concordance reliability score >0.7; (2) inadequate time interval between test and retest and use of ICC, Kappa or Concordance reliability score <0.7, adequate time interval between test and retest, Pearson/Spearman correlation >0.7; (3) an inadequate time interval between test and retest, Pearson/Spearman correlation <0.7. An additional grade was given depending on the number of participants and the level of index or correlation. A positive score (+) was given for studies with >50 participants and reliability coefficients >0.70. A negative (−) score was assigned to studies with <50 participants and reliability coefficients <0.70. Pearson and Spearman correlation were considered inadequate due to known systematic errors [57] and therefore only ICC, Kappa or Concordance were deployed in level (1) of evidence. Validity is the degree to which an instrument measures constructs [54]. The highest level of criterion validity evidence would be comparing PAQs to the gold standard—doubly labelled water (DLW) [58]. However, DLW also includes basal metabolic rate and the thermic effects of food, and therefore the use of other validated instruments is more reliable for obtaining construct validity. This is done by comparing a PAQ to another PAQ (concurrent validity), and accelerometers (criterion validity). For concurrent and criterion validity, the research team established the following levels of evidence: (1) concurrent validity score >0.8; (2) 0.8> validity score ≥0.5; (3) concurrent validity score <0.5. A positive score (+) was given for studies with >50 participants and a negative (−) score was given for studies with <50 participants.

## 3. Results

The flow of the review process is shown in Figure 1. In total, 4969 abstracts were identified, 1749 records were screened, 287 full-text articles were identified and read and 20 studies were finally included in the present review (Figure 1). The characteristics of the included studies are presented in Table 1. We included studies from 18 different EU countries, mostly from the United Kingdom (7), Spain (5) and Germany (3). Three studies were cross-national [33,59,60]. Table 1 represents information from all 20 studies included in the present review of selected PAQs [33,35,46,59,60,61,62,63,64,65,66,67,68,69,70,71,72,73,74,75], including the country where the study was carried out, the sample size, participants’ age and gender, sample description, modes and means of administration of selected studies.

Altogether, 5997 people in 23 different sub studies participated. The age range of included participants in all studies was between 18 and 75 years. In 18 out of 20 studies, the gender proportion of participants was included, whereas in two studies, gender proportion was unknown [75,76]. Regarding sampling procedures, 13 studies used convenient sample (65%), 4 random sampling (20%), 1 quota sampling (5%), 1 multistage stratified probability sampling (5%) and one study did not report a sample description [61]. Most of the studies (*n* = 13) used a self-administered mode of administration, 4 used an interview and 2 used telephone interviews. In one study, both self-administered questionnaires and an interview mode was used. All of the included studies assessed the duration and frequency of physical activity. 

Table 2 represents information from eight studies regarding the reliability of PAQs in selected studies across the EU [33,46,64,65,68,70,72,76], including information about measurement interval, results (Pearson r, Spearman ρ, Lin’s concordance correlation and Phi coefficient) and quality ratings. Most studies assessed test-retest reliability for MPA (30), and the least test-retest reliability for MVPA (5). The information for concurrent validity was reported in seven PAQ studies across the EU [33,35,46,69,70,72,75]. Information about comparison method, measured construct, correlation coefficient results and quality ratings are shown in Table 2. Most of these studies assessed the concurrent validity for tPA (11) and the least for VPA (6). Table 2 represents information from 13 studies regarding the criterion validity of PAQs in selected studies across the EU [33,46,59,62,63,64,65,68,70,71,72,73,74], including information on the country where the study was carried out, the duration of the objective assessment, the number of valid days and minutes per day, the method for validity comparison, cut-off points, epoch length, the definition of non-wear time and measured constructs. Most studies assessed the criterion validity for VPA and tPA (both 11) for MPA, while the fewest studies assessed the criterion validity for MPA (both 9).

Based on weighted correlation means, measurement construct test-retest performed the best in construct MVPA (r_w_ = 0.74), where 3 associations (of 5) were graded with level of evidence 1 (r_w_ = 0.74) and 2 with levels of evidence 2 (r_w_ = 0.73); whereas the worst were in MPA (r_w_ = 0.40) (Table 3), where 28 of 30 associations were graded with a level of evidence of 3 (r_w_ = 0.41) and only 2 with grade 2 (r_w_ = 0.58).

Based on weighted correlation means, concurrent validity was best for VPA (r_w_ = 0.72), where 4 associations were graded with levels of evidence 1 (r_w_ = 0.82) and 5 associations with levels of evidence 2 (r_w_ = 0.62) (Table 3). Concurrent validity was the lowest for tPA (r_w_ = 0.22), where 9 associations were evaluated with levels of evidence 2 (r_w_ = 0.64) and 2 with levels of evidence 3 (r_w_ = 0.38). On the other hand, VPA showed the highest validity (r_w_ = 0.72), but it should be noted that the Egger test (−5.63) showed a significant bias between included correlations coefficients in VPA (*p* < 0.0001). Based on weighted correlation means, measurement construct performed the best for VPA (r_w_ = 0.48), where 4 associations were evaluated with a level of evidence of 2 (r_w_ = 0.64) and 7 associations with a grade of 3 (r_w_ = 0.30); the worst criterion validity was noted for MPA (r_w_ = 0.14) (Table 3), with all 9 associations graded with the level of evidence of 3. Once again, although the highest criterion validity was noted for VPA, the Egger test (−5.59) showed a significant bias between included correlations coefficients in VPA (*p* < 0.0001). Results of weighted correlation coefficients for test-retest reliability, concurrent validity and criterion validity across all included studies stratified by PA intensity are presented in Figure 2.

The Egge’s bias test [53] provided evidence for publication bias for the following measurement characteristics and PA constructs: concurrent validity VPA (bias = −5.63, 95% CI: −6.80 to −4.46, *p* < 0.0001), concurrent validity tPA (bias = −0.14, 95% CI: 6.47 to 6.20, *p* = 0.97), criterion validity VPA (bias = −5.59, 95% CI: −7.38 to −3.81, *p* < 0.0001) and criterion validity tPA (bias = −3.22, 95% CI: −6.55 to 0.11, *p* = 0.09) (Table 3). The results of the risk-of-bias assessment are shown in Table 4. The total average risk of bias of all included studies was moderate (0.43). Of the 20 studies, only two were rated as having a low risk of bias (≥67% of total score) with an average of 0.73 of the total score; 10 were rated as having a moderate risk of bias (>33 and <67% of the total score) with an average of 0.45 of the total score and 8 studies were rated as having a high risk of bias (<33% of total score) with an average of 0.32 of the total score. Only 6 studies (33%) reported power calculations to determine a sufficient sample size and only 3 studies met the assumption of randomization, which is not so important to determine the reliability and validity of questionnaires [77].

## 4. Discussion

This systematic review and meta-analysis investigated the test-retest reliability, concurrent validity and criterion validity of the three most commonly used PAQs across the EU in national language versions: IPAQ-SF, GPAQ and EHIS-PAQ. We identified 20 studies that adequately tested selected PAQs in the recent 17-year period between 2003 and 2020.

The main findings include the following: (i) IPAQ, GPAQ and EHIS-PAQ were validated for MPA, MVPA and VPA in only 10 countries across EU; (ii) the assessment of MVPA is the most relevant PA outcome, since no publication bias in any of the measurement characteristics were detected and test-retest reliability was moderately high (r_w_ = 0.74), while both criterion (r_w_ = 0.41) and concurrent validity (r_w_ = 0.72) were judged to be moderate; (iii) reporting of methods and results of the studies was rather poor, leading to a high risk of bias in 8 studies and a moderate risk of bias in 10 studies, resulting in an overall moderate risk of bias with a total score of 0.43; and (iv) the representation of different EU countries may be biased, since out of 20, 7 were from the UK, 5 from Spain, 3 from Germany, 2 from Lithuania and 1 from the other countries.

Our results revealed that MPA reached the lowest overall correlations for reliability and criterion validity (reliability r_w_ = 0.42; criterion validity r_w_ = 0.14) and MVPA reached the lowest correlations for concurrent validity (r_w_ = 0.41). VPA reached the highest overall correlations (reliability r_w_ = 0.53; concurrent validity r_w_ = 0.72; criterion validity r_w_ = 0.48), but we also found publication bias in concurrent and criterion validity for this PA construct. All measurement characteristics were moderate-to-high for MVPA (reliability r_w_ = 0.74; concurrent validity r_w_ = 0.41; criterion validity r_w_ = 0.41). Since we did not detect publication bias in any of the measurement characteristics for MVPA, we suggest the assessment of MVPA to be the most relevant PA outcome. To a larger extent, research findings indicate that MVPA in particular positively influences the health of the adult population, which also resulted in the development of recommendations for policymakers to increase the MVPA of the European population [1].

Although there is no single rule of the thumb relating to an adequate sample size, test-retest intervals and statistical analysis, academics have recommended the acceptable ratio of survey items and participants to be 1:5 [49,78], including test-retest interval between three and eight days [78] and the use of ICC and Pearson correlation coefficient [54]. Based on our qualitative rating, only 8 out of 311 PA constructs within different measurement characteristics received grade 1, 144 constructs were awarded with grade 2 and 149 with grade 3. Low qualitative ratings were mostly given because studies did not use the interclass correlation (ICC), Kappa or Concordance reliability score, but the majority of studies used the Spearman coefficient of association. We recommend researchers to use Kappa or ICC in the future, because they also take into account rater bias [79]. This is a foundation for concern, since more than half of the constructs did not satisfy the preferred recommendations for assessing the reliability and validity of PAQs, and calls for a more rigorous study design in future reliability and validity investigations.

It is promising that the reliability of investigated PAQs was found to be moderate to high (r_w_ = 0.40 to 0.74). Of even greater importance, time intervals with the exception of two studies [46,76] were within the optional range [78] of the test-retest interval and ranged mostly between three and eight days. Since the reliability of MVPA and tPA was high even in the two aforementioned studies [49,78] that used one month interval between repeated assessments, this methodological weakness [49] does not hamper the conclusions of this study.

PAQs showed low-to-moderate validity (r_w_ = 0.13 to 0.48) against measures of objectively measured PA and moderate-to-high validity against subjective measures of PA (other PAQs). Our results are comparable with previous reports [48,80] that showed the validity of PAQs to range from 0.1 to 0.50 against objective measures of PA [81]. However, it should be noted that the criterion validity was validated in only six different national versions for IPAQ (Ireland, Lithuania, Spain, Sweden, Finland and United Kingdom) and four different national versions for GPAQ (Austria, Belgium, Spain and the United Kingdom) across the EU. Results indicate differences in the validity between different versions, and therefore the remaining countries assessing PA do not even know how valid their data are. Moreover, factors explaining the variation in the validity of PAQs may relate to differences in the qualitative attributes of PAQs, such as recall period and number of items as well as heterogeneity of population. It is well documented that there are differences in the prevalence of overweight and obesity [82] and physical fitness levels between different nations and countries [83], which is the governing factor to assess PA with a questionnaire. PAQs are assessing the subjective perception of PA, which is conditioned by physical fitness. Accordingly, it is exceptional that only a few studies reported the reliability and validity of PAQ, observing differences in validity between countries and sex according to body mass index (BMI) [35,62], whereas we have not found a single study that used physical fitness as a criteria. It has been found that a high BMI can reduce accuracy of devices, such as accelerometers and heart rate monitors [84]. Additionally, PA data with self-reports seems to be over- or under-estimated among participants with higher BMI [84]. We believe one of the important factors affecting the variability of PAQs’ validity to be the different physical fitness levels of the participants, and therefore an inclusion of this control might allow for a more objective assessment of PA, as well as better international comparability of PA data. The rather low concurrent validity scores found in our study may be explained by the different recall periods in investigated PAQs. Next, objective measures of PA are less dependent on long-term variation, and can more accurately capture sporadic and intermittent behaviors [48], which results in a higher validity of measured PA constructs, but a lower criterion validity of PAQs. It was often blurred which dimension of PA a PAQ was supposed to measure, which made assessing concurrent validity sometimes impossible. Moreover, it was extremely difficult to assess whether the same or somewhat modified versions of PAQs were used in some studies, and it was not always clear whether the data were derived from a self-report questionnaire or whether the questionnaire was part of an interview [37]. Nevertheless, most of the studies enthusiastically concluded that PAQ is valid, but they did not take into account risk of bias and quality assessment. However, when we applied criteria for risk of bias and quality assessment, we found this conclusion to be over-optimistic, which is in concordance with a previous review [37].

### Limitations

There are several limitations of this study that should be acknowledged: (i) although we systematically searched five biggest databases in the field of PA twice and with different investigators, it is possible that not all relevant studies are included in the present meta-analysis; (ii) the most commonly used PAQs in the included studies were IPAQ (7) and GPAQ (6), while EHIS-PAQ was included because it is the only questionnaire that is a part of the PA surveillance system of all EU member states [40]. GPAQ uses a typical week to assess PA data; however, a typical week can be different in many European countries due to weather conditions yielding different PA levels. (iii) The season of the assessed PA was not taken into account, and therefore different results could be reported from studies since the EU has four seasons; (iv) even though the quality of each study was assessed, findings from studies of a lower quality were given no less importance than the other findings; (v) sample type might have a potential impact on the results of the study, since 13 out of 20 used convenience sampling; (vi) meta-analysis included only 17 studies, whereas the systematic review included 20 studies; (vii) coefficients of associations were reported whether or not they were significant or insignificant in initial studies, potentially leading to different results if only significant results were used; (viii) according to the PROSPERO register we left Eurobarometer out of the manuscript since we did not find any validation studies; (ix) this review includes studies from the UK, although at the time of publication, the UK is no longer a part of the EU; (x) although there exist other widely used PA questionnaires, targeting specific parts of the populations, such as Physical Activity Scale for the Elderly [85], we focused only on the questionnaires targeting the general adult population; and (xi) results of the present meta-analysis refers only to the adult population and are not necessarily valid in other populations such as the elderly, children and patients.

## 5. Conclusions

Where only self-reporting is affordable due to time limitations and resources of the large-scale PA monitoring in EU adults, assessment of MVPA with GPAQ, IPAQ-SF or EHIS-PAQ is recommended. All EU countries should validate the translated PAQs in their national settings. In the validation studies, it would be advisable to employ BMI, physical fitness indicators or objective assesments of PA as validation criteria. Lastly, in order to further improve the validity and reliability of PAQ in adults, the researchers should report the results in a standardized manner to allow for the improved quality of assessment and a lower the risk of bias.

## Figures and Tables

**Figure 1 ijerph-17-07161-f001:**
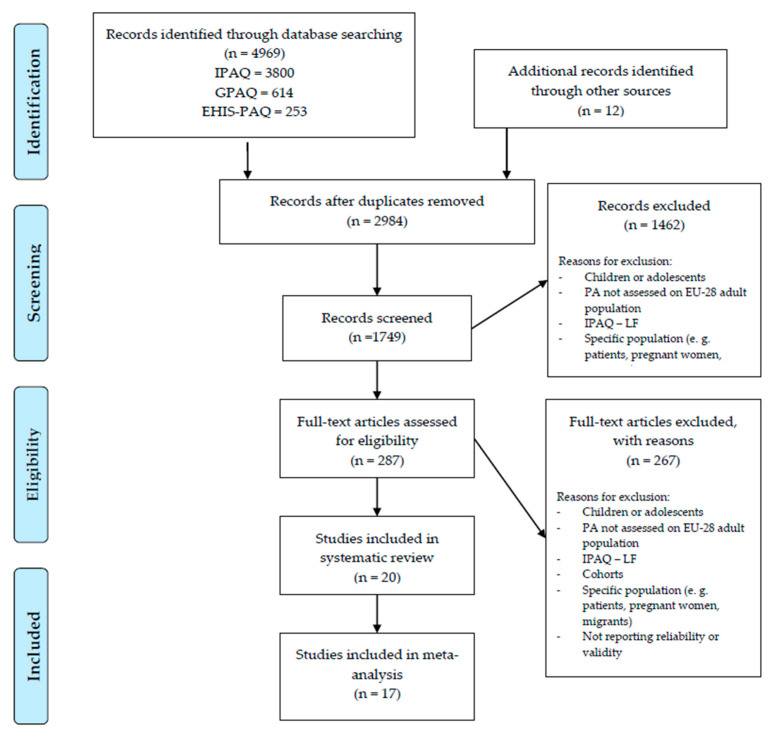
Flowchart showing the study identification process.

**Figure 2 ijerph-17-07161-f002:**
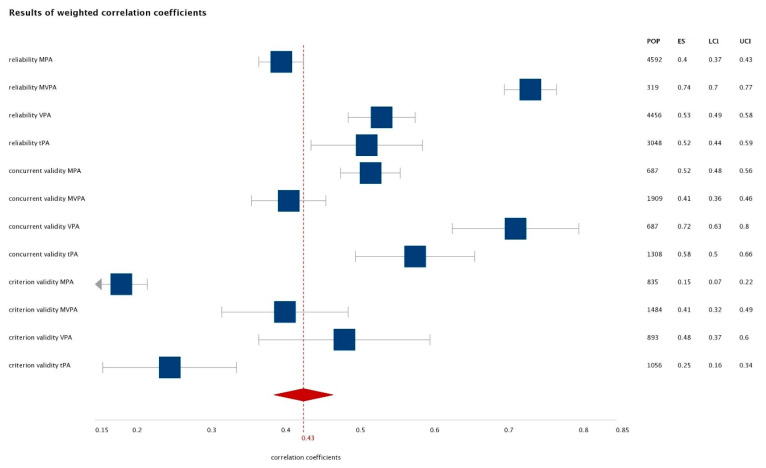
Forest plot of weighted correlation coefficients for measurement characteristics stratified by PA intensity (Note: POP—population; ESW—weighted ES; LCI—lower confidence interval; UCI—upper confidence interval).

**Table 1 ijerph-17-07161-t001:** General characteristics of selected studies of PAQs across the EU.

Author (PAQ) *Language Version*	Country	Population **				Construct					Format		
		Size	Age; (Range)	Gender (Male, *Female*)	Sample Description	Dimension	Setting	Recall Period	No. of Q	Mode and Means of Administration	Parameters	Scores	Unit of Measurement
Baumeister et al. [46] (EHIS-PAQ) *German*	DE	140	55 (18–79)	73 + *67*	Random community sample	Sitting, LPA, MPA, VPA	Work-related PA, transport, leisure time, sport activities, HEPA, sedentary	30-days	9	Self-administered Unknown mode	Duration, frequency	MVPA, LPA	Min/day, MET *min
Bull et al. [35] (GPAQ), *Portugese*	PT	67	18–75	17 + *50*	Prevalence of young participants (18–44, *n* = 56) Convenient regional sample	Sitting, MPA, VPA	Work-related PA, transport, leisure time, sedentary	7-days	19	Interview Unknown mode	Duration. frequency	VPA, MPA, TPA, sedentary	Min
Cámara et al. [61] (GPAQ), *Spanish*	ES	163	70 (67–75)	67 + *96*	Older adults from IMPACT65+ study	Sitting	Sedentary time	7-days	1	Interview Face to face	Duration, frequency	Sedentary time	Min
Cleland et al. [62] (GPAQ), *English*	UK	22	46	8 + *14*	Random national sample	Sitting, MPA, VPA	Work-related PA, transport, leisure time, sedentary	7-days	16	Self-administered Unknown mode	Duration, frequency	MVPA, sedentary	Min/day
Craig et al. [33] (IPAQ-SF), *German, English, Finnish, Dutch, Portugese, Sweedish*	Cross-national: AT, UK, FI, NL, PT, SE	2115: 200 SE1 50 SE2; 149 UK1 101 UK2 88 FI 196 PT 74 NL	47 41 35 41 56 35 33	77 + *123* 22 + *28* 68 + *81* 38 + *63* 43 + *45* 96 + *100* 34 + *40*	Specific populations Convenient samples, but collectively, the participants represented a wide range of age, education, income, and activity levels	Sitting, MPA, VPA	Leisure time PA, domestic and gardening activities, work-related PA, transport-related PA	7-days	9	Self- administered Unknown modes	Duration. frequency	Categorical measure of % min/week	Min/week
Ekelund et al. [63] (IPAQ-SF), *Sweedish*	SE	185	42 (20–69)	93 + *92*	Workers and students Convenient regional sample	Sitting, MVPA	Leisure time PA, domestic and gardening activities, work-related PA, transport-related PA	7-days	7	Telephone interview	Duration. frequency	MVPA	MET min/day. MET min/week
Kalvenas et al. [64] (IPAQ-SF), *Lithuanian*	LT	92 ^#^	18–69	reliability 29 + *63* validity 23 + *58*	Employees of university and private company Convenient sample from urban area	Sitting, MPA, VPA	Leisure time PA, domestic and gardening activities, work-related PA, transport-related PA	7-days	9	Self-administered Unknown mode	Duration. frequency	VPA, MPA+walking, MPA, WPA, sitting, TPA	
Kastelic et al. [66] (GPAQ), *Slovenian*	SI	42	M 39 F 50	37 + *5*	Crane operators and office workers Convenient sample	Sitting, MPA, VPA	Work-related PA, transport, leisure time, sedentary	7-days	16	Interview Unknown mode	Duration. frequency	sedentary	Min/day
Kleinauskiene et al. [65] (IPAQ-SF), *Lithuanian*	LT	92	18–69	29 + *63*	Convenient sample from Kaunas city	Sitting, MPA, VPA	Leisure time PA, domestic and gardening activities, work-related PA, transport-related PA	7-days	9	Self-administered	Duration. frequency	MET min/week	MET, min/week
Laeremans et al. [59] (GPAQ), *German, Spanish, English*	Cross-national: B. ES, UK	122: 41 B; 41 ES; 40 UK	35	55 + *67*	Random regional sample	Sitting, MPA, VPA	Work-related PA, transport, leisure time, sedentary	7-days	16	Self-administered Online	Duration. frequency	MPA, MVPA, VPA, sedentary	MET min/week
Milton et al. [67] (GPAQ), *English*	UK	240	18–64	119 + *121*	Quota sample from across England, Scotland and Wales	Sitting, MPA, VPA	Work-related PA, transport, leisure time, sedentary	7-days	16	Telephone interview	Duration. frequency	MVPA	Min/day
Murphy et al. [68] (IPAQ-SF), *English*	IE	155 ^##^	23	69 + 86	Students Convenient sample	Sitting, MPA, VPA	Leisure time PA, domestic and gardening activities, work-related PA, transport-related PA	7-days	9	Self-administered Unknown mode	Duration. frequency	MVPA as % in PA population	Min/week
Novak et al. [69] (GPAQ), *German*	AT	50	25	39 + *11*	Students Convenient sample	Sitting, Total PA, VPA	Work-related PA, transport, leisure time, sedentary	7-days	16	Self-administered Unknown mode	Duration. frequency	Total PA, VPA, sedentary	Min/week
Rivière et al. [70] (GPAQ), *French*	FR	87 ^###^	30	25 + *67*	Medical personnel and students, convenience sample	Sitting, MPA, VPA	Work-related PA, transport, leisure time, sedentary	7-days	16	Interview and self-administered Unknown mode	Duration. frequency	LPA, VPA, TPA, MVPA	Min/day
Rodríguez-Muńoz et. al. [74] (IPAQ)	ES	95	22	*33* *+ 62*	University students Convenience sample	Sitting, MPA, VPA	Moderate-to-vigorous PA	7-days		Self-administered Unknown mode	Duration. frequency	MVPA	Min/day
Rudolf et al. [71] (GPAQ), *German*	DE	54	28	*23* * + 31*	University students Convenience sample	MPA, VPA, Sitting	Work-related PA, transport, leisure time, sedentary	7 days	16	Self- administered Online	Duration. frequency	MPA, VPA, sedentary	Min/day
Rütten et al. [60] (IPAQ–SF), *German, Finnish, French, Italian, Dutch, Spanish, English*	Cross-national: B, FI, FR, DE, I, NL, ES, UK	951: 100 B; 127 FI; 91 FR; 223 GR; 98 I; 86 N; 128 S; 98 UK	>18	unknown	Random sample	Sitting, MPA, VPA	Leisure time PA, domestic and gardening activities, work-related PA, transport-related PA	7-days	9	Interview Face to face	Duration. frequency	VPA, MPA, sedentary	Min/week, MET
Scholes et al. [75] (IPAQ-SF), *English*	UK	1252	>16	Unknown	Multistage stratified probability sampling	Sitting, MPA, VPA	Leisure time PA, domestic and gardening activities, work-related PA, transport-related PA	7-days	9	Self-administered Pen and paper	Duration. frequency	Categorical MVPA	Min/week
Taylor et al. [72] (IPAQ-SF), *English*	UK	49	27	11 + 38	Students and university staff Convenient sample	Sitting, MPA, VPA	Leisure time PA, domestic and gardening activities, work-related PA, transport-related PA	7-days	9	Self-administered Online	Duration. frequency	MPA, MVPA	MET min/day
Vinas et al. [73] (IPAQ-SF), *Spanish*	ES	24	41	26 + *29*	Convenient sample 91% of the participants had a high level of education	Sitting, MPA, VPA	Leisure time PA, domestic and gardening activities, work-related PA, transport-related PA	7-days	9	Self-administered (Catalan version) Unknown mode	Duration. frequency		Min/day

Notes: AT—Austria; B—Belgium; D—Denmark; DE—Germany; ES—Spain; FI—Finland; FR—France; GR—Greece; I—Italy; IE—Ireland; LT—Lithuania; NL—The Netherlands; NO—Norway; PT—Portugal; SE—Sweden; SI—Slovenia; UK—United Kingdom; VPA—vigorous PA; MPA—moderate-to-vigorous PA; WPA—walking PA; TPA—total PA; LPA—light PA; * age was presented by mean or median; ** population (size, age, gender) was presented only for European country, nevertheless comparisons were made cross-national; ^#^ 92 reliability and 81 validity; ^##^ 133 reliability and 155 validity; ^###^ 68 reliability and 87 criterion validity.

**Table 2 ijerph-17-07161-t002:** Results for test-retest reliability, concurrent validity and criterion validity.

Reference (PAQ)	Study Pop	Method	Construct (Comparison Method)	Results	Rating
Baumeister et al. [46] (EHIS-PAQ)	DE	TRR	MVPA	ICC = 0.73	1
		CRV	MVPA (ActiGraph GT3X)	ICC = 0.32	3
		CCV	MVPA (IPAQ-L)	ICC = 0.45	2
			MVPA (7-d PAR)	ICC = 0.26	3
Bull et al. [35] (GPAQ)	PT	CCV	VPA (IPAQ-SF)	Spearman ρ = 0.52	2
			MPA (IPAQ-SF)	Spearman ρ = 0.50	2
			tPA (IPAQ-SF)	Spearman ρ = 0.23	3
Cleland et al. [62] (GPAQ)	UK	CRV	MVPA (ActiGraph GT3X)	Spearman ρ = 0.48	3
Craig et al. [33] (IPAQ)	SE 1	TRR	Total PA	Spearman ρ = 0.66	3
		CCV	tPA 1st session (IPAQ L7T)	Spearman ρ = 0.6	2
			tPA 2nd session (IPAQ L7T)	Spearman ρ = 0.63	2
	UK1	TRR	tPA	Spearman ρ = 0.87	2
	UK2	TRR	tPA	Spearman ρ = 0.69	3
		CRV	tPA (CSA motion detector MTI)	Spearman ρ = 0.40	3
	FI	TRR	tPA	Spearman ρ = 0.65	2
		CRV	tPA (CSA motion detector MTI)	Spearman ρ = 0.47	3
		CVV	tPA 1st session (IPAQ LUS)	Spearman ρ = 0.68	2
			tPA 2nd session (IPAQ LUS)	Spearman ρ = 0.71	2
	PT	TRR	tPA	Spearman ρ = 0.77	2
		CCV	tPA 1st session (IPAQ LUS)	Spearman ρ = 0.49	3
			tPA 2nd session (IPAQ LUS)	Spearman ρ = 0.43	3
	SE 2	TRR	tPA	Spearman ρ = 0.77	2
		CRV	tPA (CSA motion detector MTI)	Spearman ρ = 0.02	3
		CCV	tPA 1st session (IPAQ LUS)	Spearman ρ = 0.77	2
			tPA 2nd session (IPAQ LUS)	Spearman ρ = 0.87	2
	NL	TRR	tPA	Spearman ρ = 0.85	2
		CRV	tPA (CSA motion detector MTI)	Spearman ρ = 0.32	3
		CCV	tPA 1st session (IPAQ L7T)	Spearman ρ = 0.85	1
			tPA 2nd session (IPAQ L7T)	Spearman ρ = 0.88	1
Ekelund et al. [62] (IPAQ)	SE	CRV	MVPA (ActiGraph)	Pearson r = 0.17	3
			tPA (ActiGraph)	Pearson r = 0.34	3
Kalvenas et al. [64] (IPAQ)	LT	TRR	MPA (min/weak)	Spearman ρ = 0.53	3
			VPA (min/weak)	Spearman ρ = 0.67	3
			tPA (min/weak)	Spearman ρ = 0.51	3
		CRV	VPA (ActiGraph GT3X)	Spearman r = 0.40	3
			MPA (ActiGraph GT3X)	Spearman r = -0.03	3
			tPA (ActiGraph GT3X)	Spearman r = -0.11	3
Kleinauskiene [65] (IPAQ)	LT	TRR	MPA	Spearman ρ = 0.35	3
			VPA	Spearman ρ = 0.83	2
		CRV	weekly tPA 1st session	Spearman ρ = 0.27	3
			weekly tPA 2nd session	Spearman ρ = 0.06	3
Laeremans et al. [58](GPAQ)	B, ES, UK	CRV	MVPA (SWA) 1st session	Spearman r = 0.56	2
			MVPA (SWA) 1st session	Spearman r = 0.64	2
			MVPA (SWA) 1st session	Spearman r = 0.55	2
			Overall MVPA (SWA) 1st session	Spearman r = 0.54	2
			VPA (SWA) 2nd session	Spearman r = 0.62	2
			VPA (SWA) 2nd session	Spearman r = 0.69	2
			VPA (SWA) 2nd session	Spearman r = 0.59	2
			Overall VPA (SWA) 2nd session	Spearman r = 0.64	2
			MPA (SWA) 3rd session	Spearman r = 0.11	3
			MPA (SWA) 3rd session	Spearman r = 0.34	3
			MPA (SWA) 3rd session	Spearman r = 0.02	3
			Overall MPA (SWA) 3rd session	Spearman r = 0.34	3
Murphy et al. [68] (IPAQ)	IE	TRR	tPA	ICC = 0.77	2
		CRV	MVPA (ActiGraph GT1 M & GT3X)	Spearman ρ = 0.31	3
			tPA (ActiGraph GT1 M & GT3X)	Spearman ρ = 0.28	3
Novak et al. [69] (GPAQ)	AT	CCV	VPA (PAQ 24)	Spearman ρ = 0.51	2
			tPA (PAQ 24)	Spearman ρ = 0.43	3
Rivière et al. [70] (GPAQ)	FR	TRR	MPA	Spearman ρ = 0.56 ICC = 0.48	3 3
			Total VPA	Spearman ρ = 0.8 ICC = 0.84	2 1
			Total PA	Spearman ρ = 0.82 ICC = 0.58	2 2
		CRV	VPA (ActiGraph GT3X)	Spearman ρ = 0.38	3
			VPA (ActiGraph GT3X)	Spearman ρ = 0.10	3
			tPA (ActiGraph GT3X)	Spearman ρ = 0.24	3
		CCV	VPA 1st session (IPAQ-LF)	Spearman ρ = 0.86	1
			VPA 2nd session (IPAQ-LF)	Spearman ρ = 0.76	1
			MPA 1st session (IPAQ-LF)	Spearman ρ = 0.41	3
			MPA 2nd session (IPAQ-LF)	Spearman ρ = 0.58	2
			tPA 1st session (IPAQ-LF)	Spearman ρ = 0.66	2
			tPA 2nd session (IPAQ-LF)	Spearman ρ = 0.67	2
Rodríguez-Muńoz et al. [74] (IPAQ)	ES	CRV	MVPA uniaxial (Actigraph GT3x and GT3X+) male	Pearson r = 0.66	2
			MVPA uniaxial (Actigraph GT3x and GT3X+) female	Pearson r = 0.27	3
			MVPA uniaxial (Actigraph GT3x and GT3X+) all	Pearson r = 0.47	3
			MVPA triaxial (Actigraph GT3x and GT3X+) male	Pearson r = 0.65	2
			MVPA triaxial (Actigraph GT3x and GT3X+) female	Pearson r = 0.34	3
			MVPA triaxial (Actigraph GT3x and GT3X+) all	Pearson r = 0.49	3
Rudolf et al. [71] (GPAQ)	DE	CRV	MPA (ActiGraph GT3X and GPAQ +)	Spearman ρ = 0.19	3
			MPA (ActiGraph GT3X and GPAQ)	Spearman ρ = 0.17	3
			VPA (ActiGraph GT3X and GPAQ +)	Spearman ρ = 0.42	3
			VPA (ActiGraph GT3X and GPAQ)	Spearman ρ = 0.31	3
Rütten et al. [60] (IPAQ)	B	TRR	MPA days	Spearman ρ = 0.37	3
			MPA total minutes	Spearman ρ = 0.39	3
			VPA days	Spearman ρ = 0.55	3
			VPA total minutes	Spearman ρ = 0.44	3
			tPA Sum MET (moderate, vigorous, walking)	Spearman ρ = 0.53	3
	FI	TRR	MPA days	Spearman ρ = 0.28	3
			MPA total minutes	Spearman ρ = 0.55	3
			VPA days	Spearman ρ = 0.48	3
			VPA total minutes	Spearman ρ = 0.59	3
			tPA Sum MET (moderate, vigorous, walking)	Spearman ρ = 0.41	3
	FR	TRR	MPA days	Spearman ρ = 0.18	3
			MPA total minutes	Spearman ρ = 0.28	3
			VPA days	Spearman ρ = 0.36	3
			VPA total minutes	Spearman ρ = 0.44	3
			tPA Sum MET (moderate, vigorous, walking)	Spearman ρ = 0.29	3
	DE	TRR	MPA days	Spearman ρ = 0.43	3
			MPA total minutes	Spearman ρ = 0.54	3
			VPA days	Spearman ρ = 0.51	3
			VPA total minutes	Spearman ρ = 0.54	3
			tPA Sum MET (moderate, vigorous, walking)	Spearman ρ = 0.39	3
	I	TRR	MPA days	Spearman ρ = 0.21	3
			MPA total minutes	Spearman ρ = 0.22	3
			VPA days	Spearman ρ = 0.41	3
			VPA total minutes	Spearman ρ = 0.53	3
			tPA Sum MET (moderate, vigorous, walking)	Spearman ρ = 0.14	3
	NL	TRR	MPA days	Spearman ρ = 0.40	3
			MPA total minutes	Spearman ρ = 0.34	3
			VPA days	Spearman ρ = 0.34	3
			VPA total minutes	Spearman ρ = 0.41	3
			tPA Sum MET (moderate, vigorous, walking)	Spearman ρ = 0.34	3
	ES	TRR	MPA days	Spearman ρ = 0.38	3
			MPA total minutes	Spearman ρ = 0.32	3
			VPA days	Spearman ρ = 0.54	3
			VPA total minutes	Spearman ρ = 0.62	3
			tPA Sum MET (moderate, vigorous, walking)	Spearman ρ = 0.58	3
	UK	TRR	MPA days	Spearman ρ = 0.25	3
			MPA total minutes	Spearman ρ = 0.43	3
			VPA days	Spearman ρ = 0.47	3
			VPA total minutes	Spearman ρ = 0.36	3
			tPA Sum MET (moderate, vigorous, walking)	Spearman ρ = 0.50	3
	All nations	TRR	MPA days	Spearman ρ = 0.36	3
			MPA total minutes	Spearman ρ = 0.39	3
			VPA days	Spearman ρ = 0.47	3
			VPA total minutes	Spearman ρ = 0.51	3
			tPA Sum MET (moderate, vigorous, walking)	Spearman ρ = 0.45	3
Scholes et al. [75] (IPAQ)	ES	CCV	MVPA (PASBAQ) male	Pearson r = 0.43	3
			MVPA (PASBAQ) female	Pearson r = 0.40	3
Taylor et al. [72] (IPAQ)	UK	TRR	MVPA minutes	Spearman ρ = 0.67 ICC = 0.7	2 1
			Mean MVPA METs	Spearman ρ = 0.79 ICC = 0.8	2 1
			MPA total minutes	Spearman ρ = 0.59 ICC = 0.57	3 2
			MPA METs	Spearman ρ = 0.61 ICC = 0.58	3 2
			VPA min	Spearman ρ = 0.71 ICC = 0.64	2 2
			VPA METs	Spearman ρ = 0.71 ICC = 0.61	2 2
		CRV	MVPA METs (ActiGraph GT3X)	Spearman ρ = 0.08	3
			MVPA minutes (ActiGraph GT3X)	Spearman ρ = 0.13	3
			VPA METs (ActiGraph GT3X)	Spearman ρ = 0.05	3
			VPA (ActiGraph GT3X)	Spearman ρ = 0.04	3
			MPA METs (ActiGraph GT3X)	Spearman ρ = 0.11	3
			MPA (ActiGraph GT3X)	Spearman ρ = 0.14	3
			tPA (ActiGraph GT3X)	Spearman ρ = 0.14	3
		CCV	MPA MET (OSWEQ)	Spearman ρ = 0.52	2
			MPA (OSWEQ)	Spearman ρ = 0.46	3
			VPA (OSWEQ)	Spearman ρ = 0.53	2
			VPA METs (OSWEQ)	Spearman ρ = 0.53	2
			MVPA (OSWEQ)	Spearman ρ = 0.56	2
			MVPA METs (OSWEQ)	Spearman ρ = 0.62	2
Vinas et al. [73] (IPAQ)	ES	CRV	VPA (ActiGraph)	Spearman r = 0.38	3
			tPA (ActiGraph)	Spearman r = 0.27	3

Notes: TRR—test retest reliability; CRV—criterion validity; CCV—concurrent validity; AT—Austria; B—Belgium; D—Denmark; DE—Germany; ES—Spain; FI—Finland; FR—France; GR—Greece; I—Italy; IE—Ireland; LT—Lithuania; NL—The Netherlands; NO—Norway; PT—Portugal; SE—Sweden; SI—Slovenia; UK—United Kingdom; VPA—vigorous PA; MVPA—moderate-to-vigorous PA; TPA—total PA; LPA—light PA.

**Table 3 ijerph-17-07161-t003:** Summary results for test-retest reliability, concurrent validity and criterion validity across all included studies stratified by PA intensity.

Measurement Characteristic	PA Construct	Sample			Population Effect				Egger’s Bias Test			Heterogeneity		
		N (k)	k	*n*	Unweighted Mean	Weighted Mean	95% CI	80% CRI	Bias	95% CI	*p*	I^2^ (%)	Q	*p*
Reliability (test-retest	MPA	5	30	4592	0.42	0.40	0.37 to 0.43	0.32 to 0.47	0.52	−0.52 to 1.54	0.34	46.34	54.05	0.00
	MVPA	2	5	319	0.74	0.74	0.70 to 0.77	0.74 to 0.74	−0.46	−3.26 to 2.34	0.77	36.45	2.93	0.57
	VPA	3	28	4456	0.57	0.53	0.49 to 0.58	0.39 to 0.67	−0.30	−2.75 to 2.14	0.81	70.41	131.16	0.00
	tPA	5	19	3048	0.55	0.52	0.44 to 0.59	0.33 to 0.71	−0.71	−4.22 to 2.80	0.70	87.52	144.28	0.00
Concurrent validity	MPA	3	9	687	0.51	0.52	0.48 to 0.56	0.52 to 0.52	−2.53	−5.56 to 0.51	0.15	59.10	5.03	0.76
	MVPA	3	6	1909	0.43	0.41	0.36 to 0.46	0.34 to 0.47	0.41	−1.92 to 2.73	0.74	52.33	14.69	0.04
	VPA	3	9	687	0.69	0.72	0.63 to 0.80	0.56 to 0.87	−5.63	−6.80 to −4.46	0.00	84.75	52.47	0.00
	tPA	8	11	1308	0.61	0.58	0.50 to 0.66	0.43 to 0.74	−0.14	−6.47 to 6.20	0.97	55.30	81.92	0.00
Criterion validity	MPA	4	11	943	0.14	0.15	0.07 to 0.22	0.06 to 0.23	−2.05	−5.88 to 1.78	0.32	47.65	15.51	0.05
	MVPA	7	15	1484	0.42	0.41	0.32 to 0.49	0.22 to 0.60	−1.70	−5.45 to 2.05	0.38	75.40	60.96	0.00
	VPA	6	11	893	0.41	0.48	0.37 to 0.60	0.26 to 0.71	−5.59	−7.38 to −3.81	0.00	82.67	57.68	0.00
	tPA	8	11	1056	0.22	0.25	0.16 to 0.34	0.09 to 0.41	−3.22	−6.55 to 0.11	0.09	66.20	29.56	0.00

Notes: N—number of studies for selected PA construct and measurement characteristics; k—number of associations for selected construct and measurement characteristics; *n*—number of participants; CI—confidence interval; CRI—credibility interval; I^2^—I index of heterogeneity; Q—chi-square test of heterogeneity; MPA—moderate PA; MVPA—moderate-to-vigorous PA; VPA—vigorous PA; tPA—total PA.

**Table 4 ijerph-17-07161-t004:** Results of the risk-of-bias assessment.

Author (Year)	Outcome	R	BC	BV	T	BM	VO	DA	RR	PC	Total
Baumeister (2016) [46]	EHIS * + −	0	0	1	1	0	1	0	1	1	5/9 (0.56)
Bull et al. (2009) [35]	GPAQ +	0	0	1	1	0	1	0	1	0	4/9 (0.44)
Cámara et al. 2020 [61]	GPAQ +	0	0	1	0	0	1	0	1	0	3/9 (0.33)
Cleland et al. (2014) [62]	GPAQ −	1	0	1	1	0	1	1	1	1	7/9 (0.78)
Craig et al. (2003) [33]	IPAQ * + −	0	0	0	0	0	1	0	1	0	2/9 (0.22)
Ekelund et al. (2005) [63]	IPAQ −	1	0	0	1	0	1	0	0	0	3/9 (0.33)
Kalvenas et al. (2016) [64]	IPAQ * −	0	0	0	1	0	1	0	1	0	3/9 (0.33)
Kastelic et al. (2019) [66]	GPAQ −	0	0	0	1	0	1	0	1	0	3/9 (0.33)
Kleinauskienė (2012) [65]	IPAQ * −	0	0	0	1	0	1	0	1	0	3/9 (0.33)
Laeremans et al. (2016) [59]	GPAQ −	0	0	1	1	0	1	0	0	0	3/9 (0.33)
Milton et al. (2009) [67]	GPAQ +	0	0	1	1	0	1	0	1	0	4/9 (0.44)
Murphy et al. (2017) [68]	IPAQ * −	0	0	1	1	0	1	0	1	0	4/9 (0.44)
Novak et al. (2020) [69]	GPAQ +	0	0	0	1	0	1	0	1	1	4/9 (0.44)
Rivière et al. (2016) [64]	GPAQ * + −	0	0	0	1	0	1	0	1	1	4/9 (0.44)
Rodríguez-Muńoz et. al. (2020) [74]	IPAQ −	0	0	1	1	0	1	0	1	0	4/9 (0.44)
Rudolf et al. (2020) [71]	GPAQ −	0	0	0	1	0	1	1	1	0	4/9 (0.44)
Rütten et al. (2003) [60]	IPAQ *	1	0	1	1	0	1	0	1	1	6/9 (0.67)
Scholes et al. (2016) [75]	IPAQ +	1	0	0	1	0	1	0	1	0	4/9 (0.44)
Taylor et al. (2013) [72]	IPAQ * + −	0	0	0	1	0	1	0	1	1	4/9 (0.44)
Vinas (2012) [73]	IPAQ −	0	0	0	1	0	1	0	1	0	3/9 (0.33)
average of all studies		0.20	0.00	0.45	0.90	0.00	1.00	0.10	0.90	0.30	0.43

R—randomization; BC—Baseline comparable; BV—Baseline values accounted for in analyses; T—timing; BM—blinding of measures; VO—validated outcome measures; DA—dropout analysis; RR—reporting of results; PC—power calculation; Total—total score of the risk of bias (decimal format); * outcome for test-retest reliability; + outcome for concurrent validity; − outcome for criterion validity.

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
