# Peer review of "Validity and Reliability of International Physical Activity Questionnaires for Adults across EU Countries: Systematic Review and Meta Analysis"

_ijerph, 2020, doi:10.3390/ijerph17197161_

Round 1

Reviewer 1 Report

I found this paper interesting and comprehensive, detailed information of my assessment is presented below.

1. Originality/Novelty: The study question (“What is the reliability, concurrent validity and criterion validity of the three most commonly used PA questionnaires in national language versions of EU-28?”) seems original and is well defined. Moreover, it aims at comparing different PAQs and different constructs thereof used in EU-28. Both the use in European population and comparison of different PAQs seems to contribute to current knowledge.

2. Significance: As far as I understand, the results are interpreted appropriately. However, I believe that the number of databases searched is rather limited and this has been discussed as a study limitation. Moreover, the representation of different European countries in the study population seems to be a bit biased (of the 19 included studies, 7 were from UK, 4 from Spain and 3 from Germany), it is relevant to discuss this further. Also, the potential impact of samples types included studies is relevant to discuss (13 of the 19 included studies used convenient samples). Finally, I cannot see that the last sentences in the conclusion section (rows 354-358) are reflected in the study result.

3. Quality of Presentation: I believe that the article is written in an appropriate way. My knowledge in grading evidence is very limited and therefore I could not assess the appropriateness of the presentation of data and analysis. However, I found the presentation comprehensible and easy to follow.

4. Scientific Soundness: As described above, I found the research question, overall design and method scientifically sound. However, I cannot, due to own limited knowledge in the area, give a more thorough assessment of the used methods and techniques. I have described my views of search strategy and included sample in the synthesis above.

5. Appropriate and adequate references to related and previous work:Adequate references to related work seem to be used. However, I believe that a method reference should be added to row 165. The argumentation for the study’s relevance on rows 59-67 could be strengthened by references. Moreover, I cannot find a citation to reference 75 (Sember, 2020) in the text. It is now only found on in the reference list (rows 576-577).

6. English correct and readable: The English language seems appropriate and understandable. However, I believe that minor language should be corrected: For example, I suggest that” the most common” on rows 2 and 15 is changed to “most commonly used”. Also, spelling needs to be checked, for example on “preferred” (row 60),

7. Interest to the Readers:  The studied question, methods, results and methods are relevant for public health research. The results are also interesting for policy makers in public health. Therefore, I believe that the conclusions are interesting for the readership of the IJERPH.

8. Overall Merit: Increasing levels PA in populations urgent globally. Effective interventions need to be based on evidence and measurement of PA is important. This systematic review of reliability and validity of commonly used PAQs in EU contributes with valuable knowledge methods of instruments used today and identifies knowledge gaps. I believe that there is an overall benefit to publishing this work.

Author Response

We uploaded a WORD file. Please see attachment. 

Reviewer 2 Report

I congratulate the authors for conducting such an interesting study. However, numerous minor problems regarding the research were observed.

Abstract: please use the full name for EU-28, IPAQ-SF, GPAQ and EHIS-PAQ.

Line 54 – need fort he = for the

Line 84 – there is no such thing as long version of IPAQ – suggest to replace with the original IPAQ, same for line 87.

Please state the validity and reliability of the three questionnaires for all the EU countries.

Until finishing the introduction, still don’t know what is EU-28, can the authors write a paragraph on this? Besides, UK is no more in the EU since 2020. Please justify.

The literature review was lacking in the conceptual framework, the introduction of the three questionnaires and the physical measure such as the accelerometers was weak. Please add more review.

Line 105 – why didn’t use Google Scholars and Mendeley in the initial search?

Line 107 – please justify why April and May 2018 and through April and May 2020 only? The search strategy needs to more systematic and non-bias, and not already “pre-selected”.

Line 129-131 – why need to suddenly explain reliability here?

Figure 1, include the reasons for excluded papers in point format inside boxes: “Record excluded (n=1462)” and “Full-text articles excluded with reasons (n=264)”.

Since participants were European adult population, please state their occupation as well, which is an important factor to PA.

Any software was used to plot the forest plot, if yes, please list out the software name under “statistical analysis”.

For limitation, it needs further explanations because the EU’s has four seasons. It could be comparing winter with Summer for PA, and thus it is different.

In conclusion, I think it is unethical to mention that “the quality of most studies was poor”. There are not enough judgement/results to conclude this. Please restructure it.

All the best.  

Thank you. 

Author Response

(The authors gave the same response as above.)

Reviewer 3 Report

This manuscript is a review which evaluates reliability, concurrent validity, and criterion validity of three different questionnaires assessing physical activity (PA) levels. The review focuses on studies carried out in one of the European Union’s countries.

While objective measures seem to be more reliable when it comes to monitor PA, the use of subjective assessment through questionnaires are still often preferred given certain limitations associated with the use of accelerometers (e.g., higher costs). Considering the above, the current work might be a topic of general interest for scientific community in this field. Under my point of view, while the manuscript gathers some strengths such the spotless adherence to PRISMA guidelines; there are some important limitations which should be addressed.  

There is one critical point I am especially concerned about, and this is the lack of a strong and evidence-based rationale to focus on the three selected questionnaires. Given the existence of many other tools assessing PA (Sattler et al., 2020; Sylvia, Bernstein, Hubbard, Keating, & Anderson, 2014), authors should make clear why the review specifically addresses the evaluation of IPAQ-SF, GPAQ and EHIS-PAQ. Can it be proven that these are the most used questionnaires?  

On the other hand, there are some other weaknesses which should be pointed out. Firstly, some existing studies have not been included in the review despite meeting the eligibility criteria, (e.g., Rodríguez-Muñoz, Corella, Abarca-Sos, & Zaragoza, 2017). It could lead us to think that the search strategy might not be comprehensive enough to identify the existing literature addressing the research question in this work.  Secondly, there are some instances in the manuscript in which references are not used in a precise and adequate way (specific examples will be provided below).

Some more specific comments about each section are provided below. I hope the authors will find them useful.

Comments

Introduction

Under my point of view, the introduction is easy to follow although. Many of the references supporting the ideas presented in this section might be a bit outdated though.

Some specific comments are also provided aiming to help authors to improve this section.

  1. 35: Considering that the review focuses on adult population, it would be more consistent to include references about this population instead of about children and teenagers.
  2. 35-36: Some reference should be provided to support the claim “Creating optimal policies and planning effective interventions aimed at increasing PA is not possible without reliable data on the prevalence of physical inactivity.”
  3. 40: What do authors refer to with “all PA dimensions”? Some clarification is needed here.
  4. 42: Again, it would make more sense that references supporting this idea were about adults and not about children/teenagers
  5. 44: Some reference should be provided to support the claim “Large PA surveillance systems have, until recently, relied solely on PA questionnaires (PAQs) as one of the subjective assessment of PA”
  6. 58: It is not clear how references 28-30 support the idea that PA questionnaires prevail over objective measures.
  7. 65-67: Since there are previous reviews on the topic, what do authors think this new review can add to the existing findings? Do authors expect results to be different from those studies which were not exclusively focused on European population?
  8. 68-76: As mentioned before, the rationale to choose the three questionnaires which the review is about is not clear.
  9. 76; It is my belief that there might be a mistake in the reference cited since it is about US and not about EU.

Materials and Methods

This section is described in detail and covers the expected subsections for a review. In my opinion, it is a strength of the manuscript how authors carefully stick to PROSPERO guidelines. However, there are certain limitations which must be pointed out

According to the register in PROSPERO, authors aimed to include four questionnaires instead of three (IPAQ, GPAQ, EUROBAROMETER and EHIS-PAQ). Why did the authors decide to leave EUROBAROMETER out in the manuscript? Again, what was the rationale to choose these four questionnaires in the beginning of the process?

As mentioned before, I think the search strategy used might have led authors to miss some works which meet the eligibility criteria (e.g., Pearce et al., 2020; Rodríguez-Muñoz et al., 2017; Rudolf, Lammer, Stassen, Froböse, & Schaller, 2020)  

  1. 116: What does HEPA stand for? It is necessary to make it clear the first time it is used.
  2. 126: Authors treat concurrent and criterion validity as independent constructs. However, concurrent validity is actually a specific form of criterion validity (Dane, 2011). If authors are also including studies reporting on other forms of criterion validity (e.g., predictive validity), there is no need to add the specification of concurrent validity. If specifically reporting concurrent validity is an eligibility criteria, it must be clarified that studies have only been included if this type of criterion validity has been tested.
  3. 136: It would be more accurate to name this section “Quality and risk of bias assessment” since both constructs are addressed.
  4. 165-183: The authors present the procedure to assess reliability and validity in the studies exhaustively, well done. For clarity reasons, I would recommend the authors to consider the presentation of this information in a Table.

Reliability

1

Adequate time between…

2

3

Validity

1

2

3

Results

Both flow chart and table 1 are presented in a clear way. Table 2 reports information gathered from 14 studies. While the 19 studies included in the review are cited in lines 208-221, it is not clear why there are five references missing in the Table 2 (56, 62, 63, 71, 73). Please, consider what I mentioned earlier about criterion and concurrent validity. Since concurrent validity is a type of criterion validity, different codes should be used to identify method according to the changes made in the methods section.

In line 202, I think the authors mean studies when they refer to interventions.  

I would like to congratulate the authors for the design and inclusion of Table 3 which summarizes the most relevant information for the study purpose. I find it difficult though to see the association between data reported in Table 2, lines 227-244 and Table 3. For instance, according to lines 227-229, I would understand that there were 5 studies reporting test-retest reliability in the MVPA construct. However, when looking at Table 2, I would say that only the study by Baumeister et al. reports that association. In the same way, MPA test-retest reliability is suggested to be reported in 30 studies or sub-studies but it does not correspond with Table 2.

In my opinion, Table 2, Table 3 and wording in the text of the results section must be carefully revised to make sure that they are consistent. It should be clarified which constructs are being used for the calculations of weighted correlation means when more than a measure is provided for the same construct (for instance, when studies report both days and minutes of MPA).

Figure 2 and Table 4 are well done and clearly presented. Acronyms description should be included for Figure 2.

Discussion

In my opinion this is a weak section in the manuscript.

The authors report main findings, some of which can hardly be safely concluded considering the previously reported information. For instance, the authors claim that “GPAQ and IPAQ-SF are one of the most widely used international PAQs in EU” (lines 275-276). This cannot be a finding from the present work since the search strategy explicitly indicated the name of the three questionnaires under study.  Furthermore, the reference cited after this statement does not either support this idea. In the line 278 the authors claim that MVPA is the most relevant PA outcome. I do not think “relevant” is the most suitable adjective for what the authors mean. On the other hand, information provided between lines 283 and 287 is not consistent with Figure 2 (for instance, authors state that “MPA reached lowest overall correlations for all measurement characteristics (reliability rw=0.42; concurrent validity rw=0.51; criterion validity rw=0.13)”, but MVPA reached a lower correlation for concurrent validity).

The discussion of the findings in this section is done in a rather superficial way. When discussing qualitative ratings, for example, it would be interesting to further develop why constructs under study did not satisfy preferred recommendations.

As I see it, given the nature of the study, the discussion should include some recommendations for future research assessing PA, paying special attention to literature gaps and/or redundant limitations (e.g., Are there any countries in which validation studies would be recommended?, which questionnaires and/or constructs are reliable enough to be used for future research?, should the scientific community keep relying on self-report measures rather than objectively assessments?, how can future study improve the quality assessments and lower risk of bias?...)

References

It would be advisable to mark the 19 references corresponding to the studies included in the review (with an asterisk, for instance).

Dane, F. C. (2011). Evaluating research: methodology for people who need to read research. United States of America: SAGE.

Pearce, M., Strain, T., Kim, Y., Sharp, S. J., Westgate, K., Wijndaele, K., . . . Brage, S. (2020). Estimating physical activity from self-reported behaviours in large-scale population studies using network harmonisation: findings from UK Biobank and associations with disease outcomes. International Journal of Behavioral Nutrition and Physical Activity, 17(1), 40. doi: 10.1186/s12966-020-00937-4

Rodríguez-Muñoz, S., Corella, C., Abarca-Sos, A., & Zaragoza, J. (2017). Validation of three short physical activity questionnaires with accelerometers among university students in Spain. J Sports Med Phys Fitness, 57(12), 1660-1668. doi: 10.23736/s0022-4707.17.06665-8

Rudolf, K., Lammer, F., Stassen, G., Froböse, I., & Schaller, A. (2020). Show cards of the Global Physical Activity Questionnaire (GPAQ) – do they impact validity? A crossover study. BMC public health, 20(1), 223. doi: 10.1186/s12889-020-8312-x

Sattler, M. C., Jaunig, J., Tösch, C., Watson, E. D., Mokkink, L. B., Dietz, P., & van Poppel, M. N. M. (2020). Current Evidence of Measurement Properties of Physical Activity Questionnaires for Older Adults: An Updated Systematic Review. Sports Medicine, 50(7), 1271-1315. doi: 10.1007/s40279-020-01268-x

Sylvia, L. G., Bernstein, E. E., Hubbard, J. L., Keating, L., & Anderson, E. J. (2014). Practical guide to measuring physical activity. Journal of the Academy of Nutrition and Dietetics, 114(2), 199-208. doi: 10.1016/j.jand.2013.09.018

Author Response

We uploaded a WORD file. Please see the attachment. 

Reviewer 4 Report

Attached file with comments and suggestions

Author Response

(The authors gave the same response as above.)

Round 2

Reviewer 3 Report

I think the authors have done a great job addressing most of the comments provided in the first review. References have been updated and carefully revised and a new study has been included in the review. I am also happy with the authors’ response to comments about the methods and the results.

However, I still think there is  lack of a strong and evidence-based rationale to focus on the three selected questionnaires given the existence of many other tools assessing PA (Sattler et al., 2020; Sylvia, Bernstein, Hubbard, Keating, & Anderson, 2014). In the light of the authors’ response, “…it focuses on PAQ developed specifically for trans-national surveillance of PA, with the aim of generating comparable estimates across countries… “,  “…we report on these three PAQs since they are used in past and on-going international PA surveillance studies”, can thus be assumed that these are the only PAQs that have been used for PA surveillance studies? Can we safely state that if the study was replicated, the same questionnaires would be identified as the target ones?

As far as I can understand there is no rationale explaining why authors specifically examine these questionnaires and it is my belief that the study lacks the expected consistency a review should meet.

As for the discussion, while the authors have added some valuable comments with respect to the interpretation of quality and risk of bias assessment and the limitations, I believe that it should be improved by incorporating some recommendations for future research assessing PA, paying special attention to literature gaps and/or redundant limitations (e.g., Are there any countries in which validation studies would be recommended?, which questionnaires and/or constructs are reliable enough to be used for future research?, should the scientific community keep relying on self-report measures rather than objectively assessments?, how can future study improve the quality assessments and lower risk of bias?...)
